# The Effect of *Camellia oleifera* Cake Polysaccharides on Growth Performance, Carcass Traits, Meat Quality, Blood Profile, and Caecum Microorganisms in Yellow Broilers

**DOI:** 10.3390/ani10020266

**Published:** 2020-02-07

**Authors:** Jing Wang, Mengyu Zhang, Zhongyong Gou, Shouqun Jiang, Yingzhong Zhang, Minghuai Wang, Xuxiao Tang, Baohua Xu

**Affiliations:** 1Guangdong Provincial Key Laboratory of Silviculture, Protection and Utilization, Guangdong Academy of Forestry, Guangzhou 510520, China; wangjing@hotmail.com (J.W.); rainbow84397520@163.com (M.Z.); wmhgdf@163.com (M.W.); tangxuxiao666@163.com (X.T.); baohuaxu@aliyun.com (B.X.); 2Institute of Animal Science, Guangdong Academy of Agricultural Sciences, State Key Laboratory of Livestock and Poultry Breeding, Key Laboratory of Animal Nutrition and Feed Science in South China, Ministry of Agriculture and Rural Affairs, Guangdong Public Laboratory of Animal Breeding and Nutrition, Guangdong Key Laboratory of Animal Breeding and Nutrition, Guangzhou 510640, China; yozhgo917@163.com (Z.G.); jsqun3100@hotmail.com (S.J.)

**Keywords:** *Camellia oleifera* cake polysaccharides, yellow broilers, growth performance, carcass traits, meat quality, blood profile, caecum microorganisms

## Abstract

**Simple Summary:**

Plant originated polysaccharides used as feed additives have gradually become popular for the health and nutrition of broilers. In the present study, polysaccharides from *Camellia oleifera* cake (CCP) were added to the daily diet of yellow broilers. Our results indicated that the immunity of the broilers was boosted due to the increasing weight or index of gizzard, spleen, and the thymus. CCP treatment improved the juiciness and changed the meat color of broilers by increasing the cooking loss and the yellowness. Additionally, the structure of intestinal flora altered as a response to the CCP supplementation, which favored the health of broilers. Results have revealed that CCP has potential and development value as a new type of feed additive for broilers.

**Abstract:**

The study was carried out to evaluate the influence of polysaccharides from *Camellia oleifera* cake (CCP) in Lingnan yellow broilers diet from 1 to 50 days. Growth performance, carcass traits, meat quality, blood profile, and caecum microorganisms were characterized by three different levels of 0, 200 and 800 mg/kg CCP supplementation. Dietary treatment did not affect the productive trait from 1 to 50 days of age, except that average daily feed intake decreased at 42 days of age (*p* < 0.05). Additionally, the effects of CCP on various organs were different. The weight (*p* < 0.01) and index (*p* < 0.05) of bursa of Fabricius gradually decreased with the higher CCP supplementation at 21 days of the broilers diet. The gizzard weights were all higher when the broilers were fed with higher CCP concentration at 21, 42, and 50 days, respectively (*p* < 0.05). The weight and index of the spleen increased most with low CCP concentration (200 mg/kg) at 42 and 50 days. Moreover, CCP addition had no significant effect on meat quality except cooking loss (*P* < 0.05) and yellowness of meat color (*p* < 0.05). In the study of blood metabolism at 50 days of broilers, the concentration of calcium (*p* < 0.01), total cholesterol (*p* < 0.05) and uric acid (*p* < 0.01) decreased with higher CCP supplementation. CCP increased the albumin concentration (*p* < 0.001) that was highest at 200 mg/kg CCP supplementation. The addition of CCP increased the number of *Lactobacillus* and *Enterococcus faecalis* (*p* < 0.01) in the caecum of broilers, and had the potential to inhibit the growth of *Escherichia coli* (*p* = 0.11). Results showed that CCP played a role in improving intestinal flora and the immunity of yellow broilers.

## 1. Introduction

*Camellia oleifera* Abel, an evergreen oil plant, is mainly distributed in hilly regions of Southern and Central China [1]. The *Camellia* seed oil rich in unsaturated fatty acids has high nutritional value for human health [2,3]. Because of the yield of the plant edible oil requirement increasing in the recent year, the *C. oleifera* seed cake as a byproduct of seed oil production is up to 800,000 tons per year [4], which contains diverse bioactive components like polysaccharides, saponins, protein, and polyphenols [5]. The extract of *C. oleifera* seed cake has many biological functions, such as antifungal effects [6], hemolytic activity [7], slight protection as an intestinal barrier [8], as well as treating broilers against infection of *Escherichia coli* and *Staphylococcus aureus* [4].

Polysaccharides have been gradually recognized to have various biological functions, such as antidiabetes [9], antifatigue [10], antimicrobial [11], antitumor [12,13,14,15], antioxidant [16,17,18], hepatoprotection [19], hypolipidemic activity [20], immunomodulation [21,22,23], and neuroprotection [24]. Some studies reported that the *C. oleifera* cake polysaccharides (CCP) had antioxidant, antitumor [25,26,27], and hypoglycemic activity [28] that made it a suitable candidate for animal feeding.

In recent years, the productivity of the broiler industry has profoundly changed [29]. In particular, the feed accounts for approximately two-thirds of the cost of chicken breeding [30]. Lingnan yellow broilers are a kind of special chicken in China and have been selected for many years because of the relatively higher feed efficiency and meat yield. Seeking the appropriate feed additives is a growing need to enhance the health of broilers and furthering the reduction in breeding costs [31]. For yellow broilers, on the premise of guaranteeing meat flavor, it is of great significance to rationally prepare feed and improve immune capacity. Hence, the formulation of diets with a polysaccharides profile is a critical step to save the cost or improve the health of broilers that are characterized by a rapid growth rate.

The effects of dietary supplementation with CCP feeding in broilers have not been evaluated yet. Hence, our study aimed to systematically elucidate the influence of CCP on growth performance, carcass traits, meat quality, blood profiles, and caecum microorganisms of yellow broilers.

## 2. Materials and Methods

### 2.1. Crude Polysaccharides Extraction

De-oiled *Camellia oleifera* cake was provided by the Guangdong Academy of Forestry (Guangzhou, China) and powdered with a pulverizer. After the powder was extracted with distilled water (1:15, *w*/*v*) in a blender at 100 rpm, at 70 °C for 1.5 h, the liquor was centrifuged at 6000 rpm for 10 min. The supernatant was concentrated by rotavapor and subsided with ethanol (1:1, *v/v*) to remove saponin at 4 °C for 24 h. The solution was centrifuged at 6000 rpm for 10 min. The precipitate was lyophilized by vacuum refrigeration and powdered with a pulverizer. The powder was filtrated with a 60-mesh griddle and yielded crude samples. After major components analysis, the extracted samples contained 28.47% polysaccharides, 18.98% crude protein, 15.00% lignin, 2.13% cellulose, 1.50% ash, and were without tea saponin.

### 2.2. Chicks and Housing

All experimental procedures used in the current programme followed the standard practices of Lingnan yellow broilers recommended by Institute of Animal Science (GAASIAS-2018-016), Guangdong Academy of Agricultural Sciences (Guangzhou, China). The approval number of the Ethics Committee for this research is GAASIAS-2018-016. A total of 288 one-day-old chicks (Lingnan yellow broiler, a Chinese quality meat-type chicken, average 38.9 g initial body weight), obtained from Institute of Animal Science, Guangdong Academy of Agricultural Sciences, were placed in a windowless, environmentally controlled room and randomly allotted to one of three dietary treatments. Each treatment had six replicate floor pens with 16 chicks (eight female and eight male) per pen on the first day with size of 1.3 × 3.5 m. The floor was covered with wood shaving as the bedding material. Room temperature of 32 °C was maintained during the first 3 days, and then the temperature was reduced gradually until reaching 26 °C at 21 days. The chicks were kept on a 24-h constant lighting schedule and allowed to consume mashed diets and water ad libitum throughout the trial.

### 2.3. Experimental Diets

The trial was a single factorial design. All treatments were arranged with the same concentrations of all nutrients except for *Camellia oleifera* cake polysaccharides (CCP). The control group received the basal diet without any CCP supplementation. The other two dietary CCP treatments were supplemented under a randomized complete block design. In this study, 200 mg/kg and 800 mg/kg CCP (0.70 g/kg and 2.81 g/kg extracted samples) were selected according to Khalaji et al. [32] and Dong et al. [8] and added in to the broilers basic feeding. CCP was mixed with maize cob meal by step amplification, then blended in the 1% vitamin-mineral premix.

The three phases of the feeding program were starter (1–21 days), grower (22–42 days) and finisher (43–50 days). Ingredient and nutrient composition of the basal diet were based on NRC (National Research Council) [33] recommendations. Feed ingredient analyses were performed by following the procedures of AOAC (Association of Official Analytical Chemist) [34]. The ingredient composition, estimated nutrient content and determined chemical values of the experimental diets are shown in Table 1.

### 2.4. Productive Performance

During the processing phases, birds and carcasses belonging to the three experimental groups were identified clearly and kept separated. Body weight (BW) of chickens, feed intake and consumption, were recorded daily per pen at the end of starter, grower, and finisher feeding phase (21, 42 and 50 days of age). Average daily weight gain (ADWG), average daily feed intake (ADFI), and feed conversion ratio (FCR) were counted from these data by the overall rearing period and cumulatively. Mortality was checked daily and dead birds were weighed to calculate the mortality percentage and to adjust FCR during the experiment.

### 2.5. Slaughtering Measurements

At the end of each feeding phase, six replicate floor pens with two chickens (one female and one male) per pen at 21, 42 and 50 days of age, representative of the average BW of each group, were selected, placed in crates overnight, and slaughtered by severing the carotid artery and jugular vein after 16 h feed withdrawal. For each experimental group, the bursa of Fabricius, gizzard, liver, proventriculus, spleen and thymus of all the broilers in the three phases were dissected and weighed.

### 2.6. Meat Quality Attributes

At 50 days, the birds from six replicate floor pens with two chickens (one female and one male) per pen were eviscerated to obtain the carcass index on a group basis by removing abdominal fat, blood, feathers, feet, head, neck, and viscera. After air-chilling, the carcass weight of each group was recorded, and abdominal fat was collected and weighed. Skinless and deboned breast muscles from the carcass were packaged and weighed, then kept at −4 °C until the time course analysis for other meat quality attributes was undertaken. The indexes of the carcass, breast muscle and abdominal fat were calculated on a group basis as the percentage of body live weight.

Breast meat samples were thawed at room temperature for 1 h and used for the measurement of ultimate pH, drip and cooking loss, shear force, and color values. Breast muscle pH was determined at 50 days using a portable pH meter (Testo-205, Testo SE & Co. KGaA, Testo-Straβe 1, Lenzkirch, Germany). The electrodes were completely embedded in the meat samples to make them fully in contact with the tissue fluid in the muscle, and the pH value was recorded after the pH meter reading was stable. The muscle was then stored in a refrigerator at 4 °C for 24 h and then measured once. Each meat sample was determined three times, and the average value was taken as the final result. In addition, drip and cooking loss, and shear force after cooking were determined using the method described by Sirri et al. [35]. The breast muscle color value was also analyzed according to the procedure of Sirri et al. [35] for lightness (L*), redness (a*), and yellowness (b*) was measured by a reflectance colorimeter (Yasuda Seiko Co., Minolta, Japan). The results were represented as the average of three independent determinations performed on the inner surface of the sample (bone side), representing the whole muscle surface.

### 2.7. Blood Collection

Before six replicate cages per treatment with four birds (two female and two male) per cage slaughtered at 50 days, blood was withdrawn from the wing vein, collected into 4 mL lithium-heparin vials and immediately centrifuged at 1000× *g* for 15 min. Plasma was transferred into a 1.5 mL labeled sterile tube and frozen at −80 °C for further analysis. The concentrations of albumin, calcium, high density lipoprotein cholesterol, low density lipoprotein cholesterol, malondialdehyde, total antioxidant capacity, uric acid and the activity of superoxide dismutase, total antioxidant in plasma were measured using related kits (Nanjing Jiancheng Institute of Bioengineering, Nanjing, China). The plasma concentration of immunoglobulin A, immunoglobulin G and immunoglobulin M were determined by ELISA kits respectively (Abcam, Shanghai, China), Plasma Newcastle disease virus antibody was determined by ELISA kit (Shanghai Qincheng Biotechnology Co., Ltd., Shanghai, China). Standard curves constructed and run on the assay microtiter plate were used for concentrations of IgA, IgG, IgM and NDV Ab.

### 2.8. Content of Caecum Microorganisms

The digesta in the caecum from six replicate cages per treatment with four birds (two female and two male) per cage were collected into RNase-free tubes, snap frozen in liquid N_2_, and stored at −80 °C. The content of caecum microorganisms were determined by the qPCR method. Quantity-PCR analysis of five representative types of bacteria of *Bifidobacterium*, *Lactobacillus*, *Enterococcus faecalis*, *Escherichia coli*, and *Bacteroides* in caecum digesta was finished by Qingdao Personal Gene Biotechnology Co., Ltd. (Qingdao, China). The DNA of the microorganisms were extracted from caecum digesta with a QIAamp DNA Stool Mini kit (QIAGEN, Shanghai, China).

Real-time PCR with a Bio-rad detection system (Bio-rad IQ5, Berkeley, CA, USA) was performed using a qPCR SYBR Green Master Mix (Vazyme, Nanjing, China). The caecum microorganisms were amplified. The primer sequences used in the measurement and their amplification efficiencies are listed in Table 2. The PCR procedure was followed by 10 μL of 2 × SYBR real-time PCR premixture, 0.4 μL of 10 μM PCR specific primer F, 0.4 μL of 10 μM PCR specific primer R, and 10.8 μL of DNA. Real-time PCR reactions were run under the conditions of 5 min at 95 °C, followed by 40 cycles of 15 s at 95 °C, 30 s at 60 °C. In the PCR run, standard curves were carried out for the respective microbe number.

### 2.9. Statistical Analysis

The pen was considered as the experimental unit for productive performance data, while the individual chicken served as the experimental unit. The data of six replicates, each with one female and one male (or two female and two male), were all used to calculate the averages and standard errors of the mean (SEM) without being blocked by sex. Prior to productive analysis, mortality was submitted to arcsine transformation. The data set were performed independently by age as one-way ANOVA using GraphPad Prism 8.0.1 software version. The differences were indicated statistically significant at *p* < 0.05, whereas 0.05 < *p* < 0.10 was accepted as representing tendencies to differences. When the effects of treatments were significant (*p* < 0.05), Tukey’s multiple comparisons tests were applied to compare the individual means. Tabulated results were expressed as means with SEM.

## 3. Results

### 3.1. Growth Performance

The dietary supplementation with *Camellia oleifera* cake polysaccharides at 0 (CCP0), 200 (CCP1) and 800 (CCP2) mg/kg treatment on growth performance is shown in Table 3. There was no difference in body weight, average daily weight gain and feed conversion ratio in three treatments (*p* > 0.05). After 43 days of chickens, CCP treatments decreased the average daily feed intake, and the effect of CCP2 was significantly different (*p* < 0.05). From 1 to 50 days, mortality was kept low in all three CCP treatments.

### 3.2. Carcass Traits

Supplementation with CCP at 0, 200, and 800 mg/kg was fed to yellow broilers in the period of 1 to 50 days of age to assess the efficacy in organ weight, as well as the evaluation of organ index that was expressed as the percentage of carcass weight (Table 4). At 21 days of age, the weight of the bursa of Fabricius decreased in CCP treatment groups (1.53 ± 0.16 g of CCP1 and 1.52 ± 0.07 g of CCP2 vs. 2.16 ± 0.14 g of CCP0, *p* < 0.01), while the index showed a significant difference (*p* < 0.05). However, at 42 and 50 of days, the bursa of Fabricius remained unaffected among three experimental groups (*p* > 0.05). With the increased CCP content, the weight of the gizzard was higher (*p* < 0.05), although the difference was not significant between CCP1 and CCP2 treatments. It was also shown that the index of gizzard increased significantly at 21 days of age (*p* < 0.05) and presented an increasing trend at 42 and 50 days (0.05 < *p* < 0.10). In addition, CCP exhibited potential to improve the liver weight of chickens, in which the supplementation of 200 mg/kg was even better than the higher supplementation of 800 mg/kg. Two CCP treatments in the diet had no significant effect on the liver index of yellow feather broilers, except at 42 days of age (*p* < 0.05). In proventriculus study, it had limited effect on either weight or index by addition of CCP as a feed supplement to broiler diets. In the beginning feeding phase (1–21 days), no significant difference was observed among the three experimental groups for spleen weight. However, the *p*-value of spleen weight was 0.03 (<0.05) at 42 days of age and 0.006 (<0.01) at 50 days, respectively. Meanwhile, the indexes of the spleen had significant differences at 42 and 50 days (*p* = 0.04 and 0.02 < 0.05, respectively). At 42 days, the thymus weight had a significant difference (*p* < 0.05) and the index had a trend difference (*p* = 0.10).

### 3.3. Meat Quality

The effects of dietary CCP supplementation on breast meat quality attributes of chickens at 50 days are presented in Table 5. CCP-fed birds exhibited higher cooking loss compared to the CCP0 group (14.19 ± 0.48% of CCP1 and 15.51 ± 0.53% of CCP2 vs. 13.90 ± 0.49% of CCP0, *p* < 0.05). It was noted that the dietary CCP treatment also significantly increased the yellowness value compared to the control (*p* < 0.05).

### 3.4. Blood Profile

The influence of different CCP treatment in the broilers blood profile is shown in Table 6. The dietary supplement of CCP greatly increased the albumin content of broilers (25.10 ± 1.87 g/L of CCP1 and 19.13 ± 1.87 g/L of CCP2 vs. 14.72 ± 1.32 g/L of CCP0, *p* < 0.001). CCP also significantly affected the calcium concentration of yellow broilers at 50 days (*p* < 0.01). There was an increasing trend of immunoglobulin M concentration in the high CCP supplementation group (800 mg/kg, 0.05 < *p* < 0.10). The plasma total cholesterol values decreased significantly by adding CCP to diets (*p* < 0.05). Furthermore, uric acid concentration tended to be lower in CCP1 (38.07 ± 2.07 mg/L) and CCP2 (38.00 ± 1.33 mg/L) compared to CCP0 group (48.39 ± 3.47 mg/L, *p* < 0.01).

### 3.5. Caecum Microbial Number

As can be seen from Table 7, dietary supplementation with CCP had an effect on caecum microbial number of broilers. There was a positive correlation between the CCP addition and the number of *Bifidobacterium* (13.07 ± 5.57 × 10^7^ cfu/g of CCP1 and 30.87 ± 10.98 × 10^7^ cfu/g of CCP2 vs. 10.00 ± 3.09 × 10^7^ cfu/g of CCP0, *p* = 0.27) and *Lactobacillus* (14.38 ± 3.46 × 10^7^ cfu/g of CCP1 and 42.15 ± 12.57 × 10^7^ cfu/g of CCP2 vs. 2.14 ± 0.28 × 10^7^ cfu/g of CCP0, *p* < 0.01) in the caecum of broilers. In the number of *Lactobacillus*, the effect was significant between CC0 and CCP2, CCP1 and CCP2 treatments. The lower CCP concentration in the diet had a better effect on the growth of *Enterococcus faecalis* of boilers (29.83 ± 9.43 × 10^6^ cfu/g of CCP1 and 14.60 ± 4.11 × 10^6^ cfu/g of CCP2 vs. 2.71 ± 0.39 × 10^6^ cfu/g of CCP0) and the difference was greatly significant (*p* < 0.01). Additionally, it was worth noting that the CCP supplementation decreased the number of *E. coli* (6.07 ± 0.84 × 10^8^ cfu/g of CCP1 and 3.79 ± 0.76 × 10^8^ cfu/g of CCP2 vs. 9.20 ± 2.76 × 10^8^ cfu/g of CCP0, *p* = 0.11). There was no difference in the number of *Bacteroides* in the caecum of yellow broilers at 50 days (*p* > 0.05).

## 4. Discussion

Some changes are noted in growth rate and efficiency of poultry due to environmental factors, whereas the genetic element is the key factor that contributes about 90% [36,37,38,39,40]. Therefore, the effects on productive performance by CCP in poultry diets were limited. The growth performance data including body weight, average daily weight gain and feed conversion ratio showed that the addition of *Camellia oleifera* cake polysaccharides at 200 mg/kg (CCP1) or 800 mg/kg (CCP2) treatment did not significantly affect yellow broilers from the beginning to the end of the trial (*p* > 0.05). After 43 days of the chickens age, both CCP treatment groups (112.83 ± 2.45 g/bird/day of CCP1 and 108.10 ± 3.09 g/bird/day of CCP2) showed a lower average daily feed intake compared to the control (CCP0, 118.16 ± 1.35 g/bird/day, *p* < 0.05). Hence, CCP supplementation had the potential to improve feed efficiency. In a previous study, Dong et al. [8] showed that the extract of the *C. oleifera* seed had slightly beneficial effects on growth performance in broiler chickens, whereas Khalaji et al. [32] reported that *Camellia* species extraction had negative effects on broiler performance due to the saponin. In this study, the mortality was as low as 1.04% of CCP0, 2.08% of CCP1, and 1.04% of CCP2 that occurred during the first week of the trial, which was consistent with the previous report [41].

Some carcass traits are related to the immune system that protects the body from the invasion of pathogens. Larger organ weight and index indicates stronger humoral and cellular immune capacity [30]. The bursa of Fabricius is a central organ of cellular immunity, producing antibodies and B lymphocytes [42]. Infectious bursal disease [43], Met-deficiency (methionine-deficiency) [44] and excess selenium [45] in the diet that chickens were fed probably led to the reduction in bursa weight. Although at the starter stage of chicks, the weight and index of bursa of Fabricius significantly decreased in CCP treatment groups compared to CCP0 (*p* < 0.05), the difference was not significant at the grower and finisher stages of chickens. In the overall period of trial, the dietary CCP supplementation improved gizzard weight significantly (*p* < 0.05). The 200 mg/kg CCP had the potential effect of increasing the liver weight of chickens, especially the index of the liver at 42 days (*p* < 0.05). The spleen is a peripheral immune organ, and the lower CCP concentration (200 mg/kg) was more conducive to improving the weight of the spleen during the housing time. The thymus is a specific central immune organ and the place where T lymphocytes differentiate and mature, contributing to humoral and cellular immunity [42]. In a study of CCP supplementation for Lingnan broilers diet, the low dosage tended to increase thymus weight and index at 42 and 50 days of age. In the present study, supplementation with CCP increased the weight or index of gizzard, spleen, and thymus compared to a control with CCP0, indicating the enhanced the immune capacity of broilers.

European Union countries have long proposed to focus on the quality of the raw meat for consumers [46]. Most high-quality meat production on appearance, flavor, juiciness, texture of meat, and nutrition has been emphasized. Dietary CCP supplementation did not elicit any significant influence on the carcass quality such as the weight and index of carcass, breast muscle, and abdominal fat. Breast meat pH, drip loss, and shear force did not differ as well by dietary CCP treatment on yellow broilers. However, CCP treatment increased the cooking loss (*p* < 0.05) that could improve the juiciness of broilers. Meat color in boneless products, as the most important visible characteristic for customers, is also an indicator of meat quality [47]. Meat color as an important attribute of meat quality is associated with lightness, redness, and yellowness [31]. In the current study, dietary CCP had the tendency of increasing the redness value, as well as the yellowness compared to CCP0 (*p* < 0.05). Redness matches acceptability at purchase and is always favored by consumers [31]. Higher value of yellowness indicates more pale meat [48] that is considered as an unpopular appearance. Collectively, the results indicated that supplementation with CCP changed the meat quality of broiler meat.

When some antioxidants are dosed, oxidative damage can be decreased and in turn this increases the immunity [49]. However, no significant changes were detected in immunoglobulin A, immunoglobulin G, high density lipoprotein cholesterol, low density lipoprotein cholesterol, malondialdehyde, Newcastle disease virus antibody, superoxide dismutase and total antioxidant capacity of chickens, in response to the dietary CCP treatment (*p* > 0.05). Interestingly, it was found that the dietary supplement of CCP increased the albumin content of broilers. Low CCP concentration had a better effect of increasing albumin content and the difference was greatly significant (*p* < 0.001). At 50 days of broilers age, the effect of CCP on calcium content was significant (*p* < 0.01). The total cholesterol values were significantly reduced with the CCP treatment (*p* < 0.05), in agreement with the previous report [32] using the *Camellia* L. plant extract in broiler diets. Compared to the control, uric acid concentration also significantly decreased with the CCP treatment (*p* < 0.01). Hence, the dietary supplementation with CCP had a function of improving some plasma biochemical parameters, which was similar to the previous studies using the extract of *C. oleifera* seed as dietary supplementation in broilers [50,51].

The complex gut microbiota has been focused on relating to intestinal health and disease [52]. With respect to caecum microflora it was concluded that a supplement of CCP up to 200 mg/kg in the diet of Lingnan broilers promoted the growth of probiotics and had the potential to inhibit the number of pathogenic bacteria. The number of *Lactobacillus* and *Enterococcus faecalis* significantly increased with the CCP treatment compared to the control (*p* < 0.01). Additionally, saponins in forms of glucosides from *C. oleifera* seeds significantly inhibited the pathogens of *Escherichia coli* [4], and the CCP supplementation was also negatively correlated with the number of *E. coli*. As the most abundant taxon in the gut microbiota of chicken [53], *Bacteroides* significantly increased in the dietary treatment of chickens with *Enterococcus faecalis* and the extract of *C. oleifera* seed [51]. However, our study showed that no significant difference was observed in the number of *Bacteroides* (*p* > 0.05). The results proved that CCP as a new feed additive improved the structure of the microflora and enhanced immunity for broilers.

## 5. Conclusions

Dietary supplementation with *Camellia oleifera* cake polysaccharides (CCP) had the potential function of increasing feed efficiency, and had positive effects on some organ weights and the blood profile, without showing any negative effect on growth performance of Lingnan yellow broilers. Additionally, CCP exhibited capacity in improving some antioxidant capacity and changing some meat quality attributes. Intestinal probiotics showed significant alterations in response to the CCP supplementation, demonstrating the role of CCP on improving the structure of intestinal flora and favored the intestinal health in yellow chickens. Therefore, the present research indicated the potentiality of CCP as a feed additive candidate in poultry production. Further studies are necessary to define and delineate the role of CCP on the metabolism of blood and the immune function of the gut microbe in broilers.

## Figures and Tables

**Table 1 animals-10-00266-t001:** Ingredient and nutrient levels of the basal diets in each feeding phase for yellow broilers (g/kg, as-fed basis unless otherwise indicated).

Item	Composition/%		
	1–21 Days	22–42 Days	43–50 Days
Ingredient			
Corn	59.00	60.30	65.30
Soybean meal, 46% CP	30.70	26.30	21.20
Corn gluten meal, 60% CP	2.73	4.46	4.51
Soybean oil	2.83	4.10	4.83
*L*-lysine hydrochloride	0.16	0.07	0.05
*DL*-Methionine	0.17	0.09	0.02
Limestone	1.13	1.01	0.93
Dicalcium phosphate	2.05	1.93	1.53
Salt	0.23	0.23	0.23
Zeolite	0	0.51	0.40
Vitamin-mineral premix ^1^	1.00	1.00	1.00
Total	100	100	100
Calculated nutrient content			
Metabolizable energy, kcal/kg	2950	3050	3150
Crude protein	21.00	20.00	18.00
Lysine	1.16	0.99	0.84
Methionine + Cystine	0.84	0.75	0.63
Calcium	1.00	0.92	0.78
Non-phytate phosphorus	0.46	0.43	0.35

^1^ The premix provided the following per kilogram of diets: vitamin A, 5650 IU; vitamin D_3_, 500 IU; vitamin E, 40 mg; vitamin K_3_, 0.5 mg; vitamin B_1_, 3.8 mg; vitamin B_2_, 4 mg; vitamin B_6_, 3.5 mg; vitamin B_12_, 0.01 mg; nicotinic acid, 42 mg; D-pantothenic acid, 10 mg; folic acid, 0.55 mg; biotin, 0.15 mg; choline chloride, 600 mg; antioxidant, 150 mg; mold inhibitor, 1400 mg; Fe (as ferrous sulfate), 80 mg; Cu (as copper sulfate), 7 mg; Zn (as zinc sulfate), 75 mg; Mn (as manganese sulfate), 60 mg; I (as potassium iodide), 0.35 mg; Se (as sodium selenite), 0.11 mg.

**Table 2 animals-10-00266-t002:** Primer pairs used to analyze gene expression and the amplification efficiencies by qRT-PCR.

Genes	Primer Sequence	Y-Intercept	Slope	R	Efficiency/%
*Bifidobacterium*	Forward: GGGTGGTAATGCCGGATG	41.52	−4.17	0.9983	73.60
	Reverse: TAAGCCATGGACTTTCACACC				
*Lactobacillus*	Forward: CATCCAGTGCAAACCTAAGAG	41.03	−4.17	0.9983	73.70
	Reverse: GATCCGCTTGCCTTCGCA				
*Enterococcus faecalis*	Forward: CCTTATTGTTAGTTGCCATATT	42.46	−4.20	0.9994	72.90
	Reverse: ACTCGTTGTACTTCCCATTGT				
*Escherichia coli*	Forward: GTTAATACCTTTGCTCATTGA	40.15	−3.94	0.9975	79.30
	Reverse: ACCAGGGTATCTAATCCTGTT				
*Bacteroides*	Forward: GGGAGCGTAGATGGATGTTTA	39.10	−3.85	0.9995	81.80
	Reverse: CGAGCCTCAATGTCAGTTGC				

**Table 3 animals-10-00266-t003:** Growth performance (± SEM ^1^) of yellow broilers fed diets containing different contents of *Camellia oleifera* cake polysaccharides from 1 to 50 days of age.

Days	Treatment ^2^			*p*-Value
	CCP0	CCP1	CCP2	
BW ^3^				
1	38.78 ± 0.03	38.79 ± 0.03	38.75 ± 0.03	0.58
21	476.75 ± 7.24	464.92 ± 8.18	472.00 ± 3.91	0.69
42	1247.75 ± 36.33	1241.08 ± 34.40	1254.17 ± 27.36	0.31
50	1594.33 ± 51.61	1565.42 ± 45.86	1556.08 ± 42.75	0.84
ADWG ^4^				
1–21	21.35 ± 0.36	21.31 ± 0.41	21.68 ± 0.19	0.69
22–42	37.24 ± 1.49	36.95 ± 1.38	37.23 ± 1.22	0.99
43–50	41.67 ± 1.83	41.20 ± 2.13	41.24 ± 2.85	0.99
1–50	31.75 ± 1.05	31.15 ± 0.94	31.62 ± 0.91	0.90
ADFI ^5^				
1–21	34.56 ± 0.44	34.50 ± 0.45	35.13 ± 0.45	0.55
22–42	84.98 ± 1.51	84.74 ± 0.75	85.02 ± 1.14	0.98
43–50	118.16 ± 1.35 ^a^	112.83 ± 2.45 ^a,b^	108.10 ± 3.09 ^b^	0.03 *
1–50	69.55 ± 0.84	68.82 ± 0.77	68.42 ± 1.01	0.69
FCR ^6^				
1–21	1.62 ± 0.03	1.63 ± 0.03	1.62 ± 0.01	0.99
22–42	2.32 ± 0.09	2.33 ± 0.09	2.31 ± 0.07	0.99
43–50	2.76 ± 0.12	2.85 ± 0.13	3.00 ± 0.17	0.48
1–50	2.22 ± 0.07	2.23 ± 0.07	2.23 ± 0.05	0.99
Mortality/%				
1–50	1.04	2.08	1.04	−−

Not significantly *p* > 0.05, * *p* < 0.05, ^a,b^ means values with different superscripts within a row differ significantly. ^1^ Standard error of means. Each value is the mean of six replicates. ^2^ CCP0: basal diet (BD); CCP1: BD containing 200 mg/kg *Camellia oleifera* cake polysaccharides (CCP) diet; CCP2: BD containing 800 mg/kg CCP diet. ^3^ Body weight, g. ^4^ Average daily weight gain, g/bird/day. ^5^ Average daily feed intake, g/bird/day. ^6^ Feed conversion ratio, feed intake (g)/body weight gain (g).

**Table 4 animals-10-00266-t004:** Organ weight (± SEM ^1^) of yellow broilers fed with diets containing different contents of *Camellia oleifera* cake polysaccharides from 1 to 50 days of age.

Item	Days	Treatment ^2^			*p*-Value	
		CCP0	CCP1	CCP2		
Bursa of Fabricius					
Weight, g	21	2.16 ± 0.14 ^a^	1.53 ± 0.16 ^b^	1.52 ± 0.07 ^b^	0.002 **	
	42	4.17 ± 0.33	4.21 ± 0.52	3.73 ± 0.37	0.67	
	50	3.22 ± 0.31	3.61 ± 0.37	3.30 ± 0.35	0.70	
Index ^3^, %	21	0.45 ± 0.03 ^a^	0.32 ± 0.03 ^b^	0.35 ± 0.02 ^b^	0.01 *	
	42	0.33 ± 0.03	0.33 ± 0.03	0.29 ± 0.03	0.57	
	50	0.21 ± 0.02	0.23 ± 0.02	0.21 ± 0.02	0.72	
Gizzard					
Weight, g	21	12.51 ± 0.45 ^b^	12.65 ± 0.45 ^a,b^	14.13 ± 0.48 ^a^	0.03 *	
	42	25.68 ± 0.50 ^b^	29.21 ± 2.39 ^a,b^	33.22 ± 2.34 ^a^	0.04 *	
	50	26.39 ± 1.46 ^b^	31.62 ± 2.06 ^a,b^	33.45 ± 2.04 ^a^	0.04 *	
Index, %	21	2.60 ± 0.12 ^b^	2.66 ± 0.08 ^a,b^	2.97 ± 0.10 ^a^	0.03 *	
	42	2.04 ± 0.09	2.27 ± 0.18	2.55 ± 0.13	0.06	
	50	1.75 ± 0.06	1.99 ± 0.14	2.14 ± 0.13	0.09	
Liver					
Weight, g	21	12.48 ± 0.49	12.58 ± 0.52	12.64 ± 0.30	0.97	
	42	29.74 ± 1.45	32.15 ± 1.54	28.16 ± 1.18	0.14	
	50	30.86 ± 0.83	33.30 ± 1.21	32.75 ± 1.36	0.31	
Index, %	21	2.59 ± 0.08	2.64 ± 0.09	2.65 ± 0.05	0.84	
	42	2.35 ± 0.08 ^a,b^	2.51 ± 0.08 ^a^	2.22 ± 0.08 ^b^	0.04 *	
	50	2.00 ± 0.04	2.09 ± 0.06	2.08 ± 0.08	0.48	
Proventriculus					
Weight, g	21	2.59 ± 0.09	2.49 ± 0.09	2.55 ± 0.08	0.72	
	42	5.27 ± 0.29	5.57 ± 0.32	5.41 ± 0.21	0.73	
	50	5.86 ± 0.35	5.80 ± 0.31	5.22 ± 0.27	0.29	
Index, %	21	0.54 ± 0.02	0.52 ± 0.02	0.53 ± 0.01	0.79	
	42	0.42 ± 0.02	0.43 ± 0.02	0.43 ± 0.01	0.73	
	50	0.38 ± 0.02	0.36 ± 0.02	0.33 ± 0.02	0.16	
Spleen					
Weight, g	21	0.68 ± 0.06	0.70 ± 0.06	0.63 ± 0.03	0.59	
	42	3.69 ± 0.27 ^a,b^	4.78 ± 0.42 ^a^	3.51 ± 0.30 ^b^	0.03 *	
	50	3.35 ± 0.24 ^b^	4.92 ± 0.45 ^a^	4.01 ± 0.24 ^a,b^	0.006 **	
Index, %	21	0.14 ± 0.01	0.15 ± 0.01	0.13 ± 0.01	0.61	
	42	0.30 ± 0.02 ^b^	0.38 ± 0.04 ^a^	0.28 ± 0.03 ^b^	0.04 *	
	50	0.22 ± 0.02 ^b^	0.29 ± 0.02 ^a^	0.25 ± 0.01 ^a,b^	0.02 *	
Thymus					
Weight, g	21	3.01 ± 0.25	2.77 ± 0.26	2.70 ± 0.21	0.64	
	42	5.88 ± 0.26 ^b^	7.75 ± 0.43 ^a^	7.08 ± 0.58 ^a,b^	0.02 *	
	50	6.16 ± 0.72	8.05 ± 0.90	6.96 ± 0.81	0.17	
Index, %	21	0.63 ± 0.05	0.59 ± 0.06	0.57 ± 0.04	0.69	
	42	0.48 ± 0.02	0.60 ± 0.04	0.56 ± 0.05	0.10	
	50	0.40 ± 0.05	0.51 ± 0.06	0.45 ± 0.06	0.39	

Not significantly *p* > 0.05, * *p* < 0.05, ** *p* < 0.01, ^a,b^ means values with different superscripts within a row differ significantly. ^1^ Standard error of means. Each value is the mean, n = 12. ^2^ CCP0: basal diet (BD); CCP1: BD containing 200 mg/kg *Camellia oleifera* cake polysaccharides (CCP) diet; CCP2: BD containing 800 mg/kg CCP diet. ^3^ Index, organ weight/carcass weight, %.

**Table 5 animals-10-00266-t005:** Effect of dietary *Camellia oleifera* cake polysaccharides supplementation on meat quality of yellow broilers (± SEM ^1^) at 50 days.

Item		Treatment ^2^			*p*-Value
		CCP0	CCP1	CCP2	
Carcass quality					
Carcass	Weight, g	1430.17 ± 45.17	1412 ± 42.87	1378.25 ± 40.66	0.69
	Index ^3^, %	89.54 ± 0.46	89.00 ± 0.29	87.78 ± 1.26	0.29
Breast muscle	Weight, g	175.99 ± 10.48	184.16 ± 5.28	174.69 ± 6.14	0.64
	Index ^4^, %	11.09 ± 0.65	11.67 ± 0.36	11.15 ± 0.37	0.65
Abdominal fat	Weight, g	36.10 ± 3.53	34.24 ± 4.31	31.11 ± 3.34	0.64
	Index ^5^, %	2.26 ± 0.21	2.17 ± 0.26	1.94 ± 0.17	0.57
Breast muscle attributes					
PH value (n = 36)		5.80 ± 0.04	5.82 ± 0.04	5.85 ± 0.04	0.67
Drip loss, %		3.74 ± 0.69	3.24 ± 0.48	2.79 ± 0.42	0.77
Cooking loss, %		13.90 ± 0.49^b^	14.19 ± 0.48 ^b^	15.51 ± 0.53 ^a^	0.04 *
Shear force, N		31.12± 1.20	30.74 ± 1.04	31.40 ± 1.09	0.92
Meat color (n = 36)	Lightness	54.95 ± 0.49	54.86 ± 0.64	55.20 ± 0.43	0.90
	Redness	13.61 ± 0.22	13.74 ± 0.23	13.86 ± 0.22	0.74
	Yellowness	11.59 ± 0.24 ^b^	13.48 ± 0.50 ^a^	12.71 ± 0.46 ^a,b^	0.01 *

Not significantly *p* > 0.05, * *p* < 0.05, ^a,b^ means values with different superscripts within a row differ significantly. ^1^ Standard error of means. Each value is the mean, n = 12. ^2^ CCP0: basal diet (BD); CCP1: BD containing 200 mg/kg *Camellia oleifera* cake polysaccharides (CCP) diet; CCP2: BD containing 800 mg/kg CCP diet. ^3^ Index, carcass weight/body weight that was shown in Table 2, %. ^4^ Index, breast muscle weight/body weight, %. ^5^ Index, abdominal fat weight/body weight, %.

**Table 6 animals-10-00266-t006:** Effect of dietary *Camellia oleifera* cake polysaccharides on the blood profile of yellow broilers (± SEM ^1^) at 50 days.

Item	Treatment ^2^			*p*-Value
	CCP0	CCP1	CCP2	
Albumin, g/L	14.72 ± 1.32 ^c^	25.10 ± 1.87 ^a^	19.13 ± 1.87 ^b^	0.0001 ***
Calcium, mmol/L	2.51 ± 0.05 ^a^	2.27 ± 0.03 ^b^	2.37 ± 0.07 ^a,b^	0.009 **
IgA, mg/L	102.29 ± 12.91	104.07 ± 17.77	119.98 ± 27.82	0.80
IgG, mg/L	453.60 ± 31.81	446.59 ± 30.37	466.30 ± 31.77	0.90
IgM, mg/L	32.49 ± 2.30	29.47 ± 1.72	40.34 ± 4.95	0.06
HDL-C, mmol/L	14.41 ± 1.67	15.16 ± 1.09	13.98 ± 0.93	0.80
LDL-C, mmol/L	1.74 ± 0.28	2.27 ± 0.4904	1.79 ± 0.31	0.50
MDA, nmol/mL	6.26 ± 1.32	5.44 ± 0.97	6.08 ± 0.84	0.86
NDV Ab, pg/mL	313.31 ± 23.12	304.33 ± 20.95	317.64 ± 18.28	0.90
SOD, U/mL	222.20 ± 23.16	225.48 ± 13.88	251.23 ± 14.16	0.45
T-AOC, U/mL	11.74 ± 1.43	10.10 ± 1.21	9.53 ± 1.04	0.43
T-CHO, mmol/L	6.74 ± 0.49 ^a^	5.37 ± 0.27 ^b^	5.20 ± 0.22 ^b^	0.01 *
Uric acid, mg/L	48.39 ± 3.47 ^a^	38.07 ± 2.07 ^b^	38.00 ± 1.33 ^b^	0.004 **

Not significantly *p* > 0.05, * *p* < 0.05, ** *p* < 0.01, *** *p* < 0.001, ^a–c^ means values with different superscripts within a row differ significantly. ^1^ Standard error of means. Each value is the mean, n = 24. ^2^ CCP0: basal diet (BD); CCP1: BD containing 200 mg/kg *Camellia oleifera* cake polysaccharides (CCP) diet; CCP2: BD containing 800 mg/kg CCP diet. IgA, immunoglobulin A; IgG, immunoglobulin G; IgM, immunoglobulin M; HDL-C, high density lipoprotein cholesterol; LDL-C, low density lipoprotein cholesterol; MDA, malondialdehyde; NDV Ab, Newcastle disease virus antibody; SOD, superoxide dismutase; T-AOC, total antioxidant capacity; T-CHO, total cholesterol.

**Table 7 animals-10-00266-t007:** Effect of CCP on caecum microorganisms of broilers (cfu/g, ± SEM ^1^) at 50 days.

Item	Treatment ^2^			*p*-Value
	CCP0	CCP1	CCP2	
*Bifidobacterium*, × 10^7^	10.00 ± 3.09	13.07 ± 5.57	30.87 ± 10.98	0.27
*Lactobacillus*, × 10^7^	2.14 ± 0.28 ^b^	14.38 ± 3.46 ^b^	42.15 ± 12.57 ^a^	0.001 **
*Enterococcus faecalis*, × 10^6^	2.71 ± 0.39 ^b^	29.83 ± 9.43 ^a^	14.60 ± 4.11 ^a,b^	0.007 **
*Escherichia coli*, × 10^8^	9.20 ± 2.76	6.07 ± 0.84	3.79 ± 0.76	0.11
*Bacteroides*, × 10^9^	5.39 ± 0.98	4.52 ± 0.90	6.09 ± 1.37	0.61

Not significantly *p* > 0.05, * *p* < 0.05, ** *p* < 0.01, ^a,b^ means values with different superscripts within a row differ significantly. ^1^ Standard error of means. Each value is the mean, n = 24. ^2^ CCP0: basal diet (BD); CCP1: BD containing 200 mg/kg *Camellia oleifera* cake polysaccharides (CCP) diet; CCP2: BD containing 800 mg/kg CCP diet.

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
