# Peer review of "The Effect of Camellia oleifera Cake Polysaccharides on Growth Performance, Carcass Traits, Meat Quality, Blood Profile, and Caecum Microorganisms in Yellow Broilers"

_animals, 2020, doi:10.3390/ani10020266_

Round 1

Reviewer 1 Report

The data need be revised again. About all the data, you use “All data were expressed as means ± standard errors of the mean (SEM)”, why in Table 3,4,5,6,7, you used “means ± (SEM or SE or SD) and SEM, which made readers confused. Moreover, in Table 3,4,5,6,7, P-value <0.05, you did not give the significant markers among three treatments, which made your data mixed.

Author Response

Response to Reviewer 1 Comments

Point 1: The data need be revised again. About all the data, you use “All data were expressed as means ± standard errors of the mean (SEM)”, why in Table 3,4,5,6,7, you used “means ± (SEM or SE or SD) and SEM, which made readers confused.

Response 1: Thanks very much for the reviewer’s professional recommendation. The SEM in the individual column has been removed shown in Table 3, 4, 5, 6, and 7.

Point 2: Moreover, in Table 3,4,5,6,7, P-value <0.05, you did not give the significant markers among three treatments, which made your data mixed. 

Response 2: Thanks again for the reviewer’s constructive suggestion. The significant markers of a-c among three treatments have been added into the Table 3, 4, 5, 6, and 7 (P-value < 0.05). And some descriptions have been shown in “2. Materials and Methods” (Line 187-189), and “3. Results” (Line 196-197, 201, 213-214, 228-229, 240, 256-257, 269-270, and 279-280).

Line 187-189: When the effects of treatments were significant (P < 0.05), Tukey’s multiple comparisons tests were applied to compare the individual means.

Line 196-197: After 43 days of chicks, CCP treatments decreased the average daily feed intake, and the effect of CCP2 was significant difference (P < 0.05).

Line 201, 228-229, 240, 256-257, 279-280: a-b means values with different superscripts within a row differ significantly.

Line 213-214: With the increased CCP content, the weight of gizzard was higher (P < 0.05), although the difference was not significant between CCP1 and CCP2 treatments.

Line 269-270: In the number of Lactobacillus, the effect was significant between CC0 and CCP2, CCP1 and CCP2 treatments.

Reviewer 2 Report

The author tried their best to improve the manuscript. there is only one major question need to be further explained.

Seems 1 cock and 1 hen (or 2 of each) was selected for analysis. may I ask how do you deal with the data. The average of two (or four birds) or if you block the data by sex before the analysis.

Author Response

Response to Reviewer 2 Comments

Point 1: Seems 1 cock and 1 hen (or 2 of each) was selected for analysis. may I ask how do you deal with the data. The average of two (or four birds) or if you block the data by sex before the analysis.

Response 1: Thanks very much for the reviewer’s question. There were six replicates, each with 1 cock and 1 hen (or 2 cocks and 2 hens). The 12 (totally 6 cocks and 6 hens) or 24 data (totally 12 cocks and 12 hens) were all used to calculate the average and SEM values without being blocked any data by sex. The expression has been added into the “2. Materials and Methods” shown in Line 182-184.

Line 182-184: The data of six replicates, each with 1 cock and 1 hen (or 2 cocks and 2 hens), were all used to calculate the averages and standard errors of the mean (SEM) without being blocked by sex.

Reviewer 3 Report

Manuscript revision ID: animals-711598 titled:

The effect of Camellia oleifera Cake Polysaccharides on Growth Performance, Carcass Traits, Meat Quality, Blood Profile, and Caecum Microorganisms in Lingnan Yellow Broilers

The authors have introduced most of the suggested corrections.

But they did not include the suggested statistical analysis in the manuscript, please do so.

Conclusion should be rewritten to improve. Please highlight the recommendations for poultry production.

Author Response

Response to Reviewer 3 Comments

Point 1: The authors have introduced most of the suggested corrections.

But they did not include the suggested statistical analysis in the manuscript, please do so.

Response 1: Thanks very much for the reviewer’s professional recommendation. I’m sorry that I failed to understand the reviewer’s question in the first reply that was “please ANOVA one-way analysis strengthened by polynomial analysis”. According to the reviewer’s opinion, the data were analyzed by polynomial analysis. However, there was no significant correlation among CCP0, CCP1, and CCP2 treatments. Therefore, the results of polynomial analysis were not added into this manuscript.

The differences among three treatments were strengthened by Tukey’s multiple comparisons test. The significant markers of a-c have been added into the Table 3, 4, 5, 6, and 7 (P-value < 0.05). Some descriptions have been shown in “2. Materials and Methods” (Line 187-189), and “3. Results” (Line 196-197, 201, 213-214, 228-229, 240, 256-257, 269-270, and 279-280).

Line 187-189: When the effects of treatments were significant (P < 0.05), Tukey’s multiple comparisons tests were applied to compare the individual means.

Line 196-197: After 43 days of chicks, CCP treatments decreased the average daily feed intake, and the effect of CCP2 was significant difference (P < 0.05).

Line 201, 228-229, 240, 256-257, 279-280: a-b means values with different superscripts within a row differ significantly.

Line 213-214: With the increased CCP content, the weight of gizzard was higher (P < 0.05), although the difference was not significant between CCP1 and CCP2 treatments.

Line 269-270: In the number of Lactobacillus, the effect was significant between CC0 and CCP2, CCP1 and CCP2 treatments.

Point 2: Conclusion should be rewritten to improve. Please highlight the recommendations for poultry production.

Response 2: Thanks again for the reviewer’s constructive suggestion. The conclusion about poultry production has been added into the 5. Conclusions (Line 363-364).

Line 363-364: Therefore, the present study indicated the potentiality of CCP as a feed additive candidate in the poultry production.

This manuscript is a resubmission of an earlier submission. The following is a list of the peer review reports and author responses from that submission.

Round 1

Reviewer 1 Report

Comments to the Author:

This manuscript was to investigate the effect of Camellia oleifera cake polysaccharides on growth performance, meat quality and so on in Lingnan Yellow Broilers. This kind of study is interesting and benefit to broiler production. However, there are many major questions.

Major questions:

About “Crude polysaccharides extraction”, You cheese the de-oiled cake to extract the polysaccharides, which is different with Dong et al.(2016) who used seed, why? Moreover, why not measure the concentration of polysaccharides in crude polysaccharides extraction? The concentration of polysaccharides determined the supplemented levels of CCP in experimental diets. About all the data, you use “All data were expressed as means ± standard errors of the mean (SEM)”, why in Table 3,4,5,6,7, you used “means ± (SEM or SE or SD) and SEM, which made readers confused. Moreover, in Table 3,4,5,6,7, P-value <0.05, you did not give the significant markers among three treatments, which made your data mixed. Table 1 you analysed the concentration of Salt in diets, please give its standard methods. In addition, please add the mixed method of CCP to experimental diets, you first mixed CCP in premix or mixed CCP with basal diet? The introduction and discussion is very thin.

Minor questions:

The title need be revised. The English style need be revised throughout the manuscript. L37-38 P=0.27 or P=0.11 can represent the effectiveness of CCP on intestinal flora? Table 1 Calcium hydrogen phosphate? We did not use the chemical name of CaHPO4 in animal nutrition. Table 1 Corn gluten meal, 60%, what’s 60%? I think it means 60% CP, but 60% CP is too high for corn gluten meal, please check. L144 The first appear Abbreviation, please give the full name. L147 gene expression for caecum microorganism? L148 The caecum tissue? Or caecum digesta or content? Please check. L161-162 it is mixed. Table 2 suggest the unit of BW use “g”. L186-187 “Data are means of 6 replicate cages per treatment with 1 hen and 1 cock per cage (n=12)” should be Data are means of 6 replicate cages per treatment with 6hen and 6cock per cage (n=6). L302-303 there was no linear relationship with….. it is not a fact based on your experimental design and your data. L304-306 I think you can’t get the conclusion based on your data, for example feed efficiency

Author Response

Response to Reviewer 1 Comments

Point 1: About “Crude polysaccharides extraction”, You cheese the de-oiled cake to extract the polysaccharides, which is different with Dong et al.(2016) who used seed, why?

Response 1: In Dong’s report, the extract was prepared by enzyme-assisted aqueous technology and the product contained 30% total glucose, 30% tea saponin, 8% crude protein and 25% crude ash. Our study used the method of water extraction and ethanol precipitation to obtain the extract that contained 28.47% polysaccharides, 18.98% crude protein, 15.00% lignin, 2.13% cellulose, 1.50% ash, and without tea saponin. Therefore, both of extraction method and component content were different between Dong’s and our studies.

Point 2: Moreover, why not measure the concentration of polysaccharides in crude polysaccharides extraction? The concentration of polysaccharides determined the supplemented levels of CCP in experimental diets.

Response 2: Thank you for reminding us. The crude polysaccharides extract was analyzed that we did not mention it in Method. The explanation has been added into 2.1 Crude polysaccharides extraction (Line 82-84). The dosage of extract has also been described into 2.3 Experimental diets (Line 102-103).

Line 82-84: After major components analysis, the extracted samples contained 28.47% polysaccharides, 18.98% crude protein, 15.00% lignin, 2.13% cellulose, 1.50% ash, and without tea saponin.

Line 98-99: In this study, 200 mg/kg and 800 mg/kg CCP (0.70 g/kg and 2.81 g/kg extracted samples) were selected.

Point 3: About all the data, you use “All data were expressed as means ± standard errors of the mean (SEM)”, why in Table 3,4,5,6,7, you used “means ± (SEM or SE or SD) and SEM, which made readers confused.

Response 3: Standard error of means (SEM) was used to analyze the data in the whole manuscript that expressed in 2.9 Statistical analysis (Line 188-189). SEM were also shown in titles and notes of Table 3-7.

Line 188-189: Tabulated results were expressed as means with standard errors of the mean (SEM).

Line 197-198: Table 3. Growth performance (± SEM 1) of yellow broilers fed diets containing different contents of Camellia oleifera cake polysaccharides from 1 to 50 days of age.

Line 224-225: Table 4. Organ weight (± SEM 1) of yellow broilers fed with diets containing different contents of Camellia oleifera cake polysaccharides from 1 to 50 days of age.

Line 236-237: Table 5. Effect of dietary Camellia oleifera cake polysaccharides supplementation on meat quality of Lingnan yellow broilers (± SEM 1) at 50 days.

Line 253-254: Table 6. Effect of dietary Camellia oleifera cake polysaccharides on blood profile of Lingnan yellow broilers (± SEM 1) at 50 days.

Line 275-276: Table 7. Effect of CCP on caecum microorganisms of Lingnan broiler chicks (cfu/g, ± SEM 1) at 50 days.

Point 4: Moreover, in Table 3,4,5,6,7, P-value <0.05, you did not give the significant markers among three treatments, which made your data mixed.

Response 4: Significant markers such as *, **, ***, and ****, were mentioned behind each data that p-value was less than 0.05. For example, the symbol of * was marked in last column of Table 3 and 5. The symbols of * and ** were shown in P-value column in Table 4. And each note in Table 3-7 was expressed as “Not significantly P > 0.05, *P < 0.05.” Please check the P-value column and the notes in Table 3-7 (Line 199, 226, 238, 255, and 277).

Line 199 and 238: Not significantly P > 0.05, *P < 0.05.

Line 226 and 277: Not significantly P > 0.05, *P < 0.05, **P < 0.01.

Line 255: Not significantly P > 0.05, *P < 0.05, **P < 0.01, ***P < 0.001.

Point 5: Table 1 you analysed the concentration of Salt in diets, please give its standard methods.

Response 5: Thanks for your questions. Admittedly, there were some errors of presentation of nutrient contents. All the nutrient contents in diet including crude protein, calcium and salt were calculated values. We did not analyze the crude ether, crude fiber and carbohydrates in diet. In addition, we revised some inappropriate contents in the Table 1.

Point 6: In addition, please add the mixed method of CCP to experimental diets, you first mixed CCP in premix or mixed CCP with basal diet?

Response 6: Thanks for reviewer’s suggestion. We first mixed CCP with maize cob meal with method of step by step amplification, then mixed in the 1% vitamin-mineral premix. We meanwhile added this description of mixed method of CCP in the revised 2.3 Experimental diets (Line 104-105).

Line 104-105: CCP was mixed with maize cob meal by step amplification, then blended in the 1% vitamin-mineral premix.

Point 7: The introduction and discussion is very thin.

Response 7: Thanks to reviewer’s comments. The introduction and discussion has been revised and improved. Wish it would be better.

Point 8: The title need be revised.

Response 8: Thanks to reviewer’s suggestion. The title has been revised to “The effect of Camellia oleifera Cake Polysaccharides on Growth Performance, Carcass Traits, Meat Quality, Blood Profile, and Caecum Microorganisms in Lingnan Yellow Broilers” (Line 2-5).

Line 2-5: The effect of Camellia oleifera Cake Polysaccharides on Growth Performance, Carcass Traits, Meat Quality, Blood Profile, and Caecum Microorganisms in Lingnan Yellow Broilers

Point 9: The English style need be revised throughout the manuscript.

Response 9: Thanks to reviewer’s comment. This manuscript has been further modified by several experts. Wish the language had been improved.

Point 10: L37-38 P=0.27 or P=0.11 can represent the effectiveness of CCP on intestinal flora?

Response 10: The addition of CCP could increase the number of Bifidobacterium, although the difference was not significant (P=0.27). Under reviewer’s suggestion, the p-value more than 0.05 has been removed and the sentence also has been modified (Line 39-41).

Line 39-41: The addition of CCP increased the number of Lactobacillus and Enterococcus faecalis (P < 0.01) in the cecum of broilers, and had the potential to inhibit the growth of Escherichia coli (P = 0.11).

Point 11: Table 1 Calcium hydrogen phosphate? We did not use the chemical name of CaHPO4 in animal nutrition.

Response 11: Thanks for reviewer’s questions. Admittedly, there were some errors of presentation of nutrient contents. In Table 1, the chemical name of CaHPO4 has been replaced with Calcium hydrogen phosphate dihydrate to be exact.

Point 12: Table 1 Corn gluten meal, 60%, what’s 60%? I think it means 60% CP, but 60% CP is too high for corn gluten meal, please check.

Response 12: Thanks for reviewer’s professional questions. Some errors of presentation of nutrient contents was in Table 1. Yes, corn gluten meal in Table 1, 60% means its CP level and we supplemented this in Table 1. There are 2 types of corn gluten meal used in animal feeds in China, including 60% CP, 50% CP and 40% CP levels. We used the corn gluten meal with 60% CP frequently and once determined the CP content, which even can reach 63.5%. In addition, we revised some inappropriate contents in the Table 1.

Point 13: L144 The first appear Abbreviation, please give the full name.

Response 13: All abbreviation of IgA, IgG, IgM, HDL-C, LDL-C, MDA, NDV Ab, SOD, T-AOC, and T-CHO have been modified to full names of immunoglobulin A, immunoglobulin G, immunoglobulin M, high density lipoprotein cholesterol, low density lipoprotein cholesterol, malondialdehyde, Newcastle disease virus antibody, superoxide dismutase, total antioxidant capacity, total cholesterol, shown in Line 158-163.

Point 14: L147 gene expression for caecum microorganism?

Response 14: Thanks for reviewer’s professional questions. The original subtitle was incorrect and has been revised to “2.8 Content of caecum microorganisms” (Line 167).

Point 15: L148 The caecum tissue? Or caecum digesta or content? Please check. L161-162 it is mixed.

Response 15: It should be caecum digesta. Our description was not misunderstanding and has been revised to caecum digesta (Line 168-169).

Line 168-169: The digesta in caecum from 6 replicate cages per treatment with 4 birds (2 hens and 2 cocks) per cage were collected into RNase-free tubes.

Point 16: Table 2 suggest the unit of BW use “g”.

Response 16: Thanks to reviewer’s comments. The unit of BW has been revised to “g” in Table 3.

Point 17: L186-187 “Data are means of 6 replicate cages per treatment with 1 hen and 1 cock per cage (n=12)” should be Data are means of 6 replicate cages per treatment with 6hen and 6cock per cage (n=6).

Response 17: In 2.2 Chicks and housing of Material and Methods, it was designed that “Each treatment had 6 replicate floor pens with 16 chicks (8 hens and 8 cocks) per pen at first day” (Line 92-93). However, “at the end of each feeding phase, 6 replicate floor pens with 2 chicks (1 hen and 1 cock) per pen at 21, 42 and 50 days of age, representative of the average BW of each group, were selected, placed in crates overnight, and slaughtered by severing the carotid artery and jugular vein after 16 hours feed withdrawal” (Line 127-130).

16 chicks were put in each pen at first day, and 2 chicks/pen in each feeding phase of 21, 42, and 50 days were sampled and slaughtered. It was easy to cause misunderstanding, so the expression in Table 3-7 has been modified to “Each value is the mean of 6 replicates” (Line 199-200) and “Each value is the mean, n=12” (Line 226-227, 238-239, 255-256, and 277-278).

Line 199-200: Each value is the mean of 6 replicates.

Line 226-227, 238-239, 255-256, 277-278: Each value is the mean, n=12.

Point 18: L302-303 there was no linear relationship with….. it is not a fact based on your experimental design and your data.

Response 18: The conclusion was inaccurate about the linear relationship. So we deleted this sentence in Conclusions. According to the reviewer’s suggestion in Point 7, the Conclusion in this manuscript has been revised.

Point 19: L304-306 I think you can’t get the conclusion based on your data, for example feed efficiency.

Response 19: Thanks to reviewer’s suggestion. The conclusion was also inaccurate about feed efficiency. So we revised it to “Dietary supplementation with Camellia oleifera cake polysaccharides (CCP) had potential function in increasing feed efficiency” (Line 358-360). The same with the Response 18, the Conclusion in this manuscript has been revised.

Reviewer 2 Report

Summary:

The current study was conducted to evaluate the effects of camellia obleifera cake on growth performance, meat characteristics, blood profile, and immune-related response in broilers. This study involved in many parameters that are well composed to evaluate this potential feed additive. However, there are some significant errors/confusion/mistakes in statistics, results in interpretations, discussions, and conclusions. If the author did a correct statistical analysis and could better interpret how they were performed (methods and experimental units), and revised the discussion throughout. It could be considered for publication.
Here are some major questions that I have:
1. introduction needs to improve significantly: the function of C. oleifera needs to be better summarized with a specific function in different species. Based on your discussion, I don't think it has never been studied. Please summarize previous studies in chicken.
2. The individual chicken is not the experimental unit. If most of the data are analyzed using an individual bird as a statistical unit, the results were unacceptable. Please well indicated the experimental unit in each method. For example, 1 birds/pen instead of 6 birds/group. Only the replicate (pen) can be an experimental unit. What method was used for mean separation? And it should be indicated in all the tables with subscripts.
3. the conclusions and summaries of each analysis can not be interpreted with the author's bias. All the results, especially the negative ones, need to include in the summary.
4. the language needs to be improved

Comments:
Line 36-38: the p-value is more than 0.05. It is not improved. Please remove them.
Line 46: please indicated C. oleifera is a short form of camellia oleifera. And please keep consistent with using either short form or completed name for the following discussion.
Line 90: what is the inert filter?
Line 114: please indicate how many birds were used in each experimental unit (pen). 12 birds are not 12 experimental units.
Line 126: please indicate how many birds were used for this analysis
Line 142: it is plasma, not the serum. Please double check.
Line 143: please use the full name of each parameter
Line 149: please describe in details of DNA isolation procedure
Line 162: the individual chick is not an experimental unit. If this statement is applied in any of analysis, the result should be removed from this manuscript. Please double-check or re-analysis data.
Line 193: explain in detail what is the index? And what is carcass weight: live, hot, or cold?
Line 219: you can't ignore the reduction of bursa weight.
Line 243: how about the increase of yellowness? It should be included while summarizing the results.
Line 280: have you do any correlation analysis? Or you can't state there was a positive correlation. Same for line 302.

Author Response

Response to Reviewer 2 Comments

Point 1: introduction needs to improve significantly: the function of C. oleifera needs to be better summarized with a specific function in different species. Based on your discussion, I don't think it has never been studied. Please summarize previous studies in chicken.

Response 1: Thanks to the reviewer’s comments. The biological functions of Camellia oleifera has been discussed in the Introduction (Line 54-56). Yingzhong Zhang et al. (Molecule, 2018) reported the fast analysis of C. oleifera oil in different species by near infrared spectroscopy. However, regretfully, the research about the function in different species of C. oleifera could not been found yet.

Line 54-56: The extract of C. oleifera seed cake have many biological functions, such as antifungal effect, hemolytic activity, slightly protection for intestinal barrier, as well as treating broilers against infection of Escherichia coli and Staphylococcus aureus.

Point 2: The individual chicken is not the experimental unit. If most of the data are analyzed using an individual bird as a statistical unit, the results were unacceptable. Please well indicated the experimental unit in each method. For example, 1 birds/pen instead of 6 birds/group. Only the replicate (pen) can be an experimental unit. What method was used for mean separation? And it should be indicated in all the tables with subscripts.

Response 2: There were 6 replicate pens with 2 or 4 chicks in different analyzes and measurements that has been mentioned in Materials and Methods of Line 93-94, 128-129, Line 155-156, as well as in Notes of Table 3-7 (Line 202-203, 228-229, 240-241, 257-258, and 278-279).

Line 93-94: Each treatment had 6 replicate floor pens with 16 chicks (8 hens and 8 cocks) per pen at first day with size of 1.3 × 3.5 m.

Line 128-129: 6 replicate floor pens with 2 chicks (1 hen and 1 cock) per pen at 21, 42 and 50 days of age, representative of the average BW of each group, were selected.

Line 155-156: Before 6 replicate cages per treatment with 4 birds (2 hens and 2 cocks) per cage slaughtered at 50 days, blood was withdrawal from the wing vein.

Line 202-203: Each value is the mean of 6 replicates.

Line 228-229, 240-241, 257-258, and 278-279: Each value is the mean, n=12.

Point 3: the conclusions and summaries of each analysis can not be interpreted with the author's bias. All the results, especially the negative ones, need to include in the summary.

Response 3: Thanks to reviewer’s kindly suggestion. In this study, there was not negative function of Camellia oleifera cake polysaccharides. Our summary and conclusion were inaccurate because some differences were not significant. Therefore, many discussions have been revised such as in Line 22-24, Line 356-358.

Line 22-24: Results revealed that CCP had a good potential and development value as a new type of feed additive for broilers.

Line 356-358: Dietary supplementation with Camellia oleifera cake polysaccharides (CCP) had potential function in increasing feed efficiency.

Point 4: the language needs to be improved

Response 4: Thanks to reviewer’s suggestion. This manuscript has been further modified by several experts. Wish the language had been improved.

Point 5: Line 36-38: the p-value is more than 0.05. It is not improved. Please remove them.

Response 5: The addition of CCP could increase the number of Bifidobacterium, although the difference was not significant (P=0.27). Under reviewer’s suggestion, the p-value more than 0.05 has been removed and the sentence also has been revised.

Line 40-42: The addition of CCP increased the number of Lactobacillus and Enterococcus faecalis (P < 0.01) in the cecum of broilers, and had the potential to inhibit the growth of Escherichia coli (P = 0.11).

Point 6: Line 46: please indicated C. oleifera is a short form of camellia oleifera. And please keep consistent with using either short form or completed name for the following discussion.

Response 6: In formal writing, the full name (Camellia oleifera) was used for the first time, followed by the abbreviation (C. oleifera). Special notes and instructions were not required. The full name and the abbreviation have been further checked. Wish it would be correct.

Point 7: Line 90: what is the inert filter?

Response 7: Sorry, I could not understand reviewer’s comments. Inert filter was not mentioned in Line 90, even in the whole manuscript.

Point 8: Line 114: please indicate how many birds were used in each experimental unit (pen). 12 birds are not 12 experimental units.

Response 8: In 2.2 Chicks and housing of Material and Methods, it was designed that “Each treatment had 6 replicate floor pens with 16 chicks (8 hens and 8 cocks) per pen at first day” (Line 93-94). However, “at the end of each feeding phase, 6 replicate floor pens with 2 chicks (1 hen and 1 cock) per pen at 21, 42 and 50 days of age, representative of the average BW of each group, were selected, placed in crates overnight, and slaughtered by severing the carotid artery and jugular vein after 16 hours feed withdrawal” (Line 128-131).

16 chicks were put in each pen at first day, and 2 chicks/pen in each feeding phase of 21, 42, and 50 days were sampled and slaughtered. It was easy to cause misunderstanding, so the expression in Table 3-7 has been modified to “Each value is the mean of 6 replicates” (Line 202-203) and “Each value is the mean, n=12” (Line 228-229, 240-241, 257-258, and 278-279).

Line 202-203: Each value is the mean of 6 replicates.

Line 228-229, 240-241, 257-258, and 278-279: Each value is the mean, n=12.

Point 9: Line 126: please indicate how many birds were used for this analysis

Response 9:6 replicate floor pens with 2 chicks (1 hen and 1 cock) per pen” has been added into Line 134 to express the sampling method for analysis.

Line 134-135: At 50 days, the birds from 6 replicate floor pens with 2 chicks (1 hen and 1 cock) per pen were eviscerated to obtain carcass index.

Point 10: Line 142: it is plasma, not the serum. Please double check.

Response 10: Thanks for your reminding. There were all plasma samples which we gained, not serum, and we revised all of these in the revised manuscript.

Point 11: Line 143: please use the full name of each parameter

Response 11: All abbreviation of IgA, IgG, IgM, HDL-C, LDL-C, MDA, NDV Ab, SOD, T-AOC, and T-CHO have been modified to full names of immunoglobulin A, immunoglobulin G, immunoglobulin M, high density lipoprotein cholesterol, low density lipoprotein cholesterol, malondialdehyde, Newcastle disease virus antibody, superoxide dismutase, total antioxidant capacity, total cholesterol (Line 159-166).

Point 12: Line 149: please describe in details of DNA isolation procedure

Response 12: The DNA isolation is a classical method and the specific protocol is described in the QIAamp DNA Stool Mini Kit of the cecum digesta (Line 175-177). There are up to total 19 steps of protocol in the DNA kit that would be too much to describe. In our study, the isolation procedure was not changed. Therefore, it was only pointed out the information of DNA extraction kit of cecum content without describing the specific detail.

Line 175-177: DNA of microorganisms was extracted from caecum digeta with QIAamp DNA Stool Mini kit (QIAGEN, Shanghai, China).

Point 13: Line 162: the individual chick is not an experimental unit. If this statement is applied in any of analysis, the result should be removed from this manuscript. Please double-check or re-analysis data.

Response 13: The question is the same with Point 2, 8, and 9. Because the statement was not accurate, it leaded to many misunderstandings. All relevant descriptions have been revised and the explanations could be seen in the Response of Point 2, 8, and 9.

Point 14: Line 193: explain in detail what is the index? And what is carcass weight: live, hot, or cold?

Response 14: The explanation about index is mentioned in note of Table 4 (Line 235-236) without showing in paragraph.

Line 235-236: 3 Index, organ weight / carcass weight, %.

Point 15: Line 219: you can't ignore the reduction of bursa weight.

Response 15: Thanks for your suggestion. The reduction of bursa weight has been supplemented in Discussion (Line 301-305).

Line 301-305: Infectious bursal disease [43], Met-deficient [44] and excess selenium [45] in diet chicken fed probably lead to the reduction of bursa weight. Although at starter stage of chicks, the weight and index of bursa of Fabricius significantly decreased in CCP treatment groups compared to CCP0 (P < 0.05), the difference was not significant at grower and finisher stages of chicks.

Point 16: Line 243: how about the increase of yellowness? It should be included while summarizing the results.

Response 16: Thanks for your suggestion. The increase of yellowness may due to the brown camellia oleifera cake polysaccharides fed by birds. We supplemented this inferring in Discussion (Line 326-328).

Line 326-328: Higher value of yellowness indicates more pale meat [48] that is considered as unpopular appearance. Collectively, the results indicated that supplementation with CCP changed the meat quality of broiler meat.

Point 17: Line 280: have you do any correlation analysis? Or you can't state there was a positive correlation. Same for line 302.

Response 17: Thanks to reviewer’s professional comments. Supplementation with CCP could improve the growth of Lactobacillus and Enterococcus faecalis. However, the difference was not significant that CCP inhibited the growth of Escherichia coli. The description has been revised in manuscript (Line 343-346).

Line 343-346: With respect to caecum microflora it was concluded that supplement of CCP up to 200 mg/kg in the diet of Lingnan broilers promoted the growth of probiotics and had the potential to inhibit the number of pathogenic bacteria.

Reviewer 3 Report

Manuscript revision ID: animals-687339 titled:

Influence of Camellia oleifera Cake Polysaccharides on Growth Performance, Carcass Traits, Meat Quality, Blood Profile, and Caecum Microorganisms in Lingnan Yellow Broilers Diets

L21 and L39– what level of CCP is recommended by the Authors as a feed additive in broiler nutrition.

Table 1 – shows the base diet. What is the energy and polysaccharide content in experimental mixtures CCP supplemented? Please complete this information.

L75- Please provide the approval number of the Ethics Committee for this research.

L95 – Why salt, not crude ether, crude fiber and carbohydrates were determined in the mixtures?

L146 – To what reference values were the obtained results related? Please complete this information.

L165 – please ANOVA one-way analysis strengthened by polynomial analysis.

L166 – results are given as average. How many repetitions were there in individual analyzes and measurements?

L169-170 – this is not information about reporting results. They should be moved to the Discussion chapter.

L178 – The authors suggest that CCP can improve feed efficency is too far-reaching. Statistically significant differences were found only in one parameter in the last 7 days of fattening. Supplementation had no significant effect on other CCP parameters. Please provide this information more precisely.

L187 – Please specify the number of measurements. In Table 3, the authors state that measurements of broiler production parameters were carried out in the amount of n = 12 (1 hen and 1 cock), while in Material and Methods state that the study was carried out at n = 16 (8 hen and 8 cock).

L192-194, 196-198, 228-229, 241-244, 277 – please transfer this information to Discussion.

The manuscript is missing the Discussion chapter. This is a serious shortcoming in this publication. The results were presented with a very poor background of basic information. The authors did not even try to translate the results obtained or analyze the processes involved. Without this part, the manuscript cannot be published - it is not of adequate scientific value.

Conclusion

L301 – And what were the tests? Not nutritional?

Please rewrite Conclusion. What are the most important achievements of using CCP? No recommendations for poultry production. Please provide application forms.

Author Response

Response to Reviewer 3 Comments

Point 1: L21 and L39– what level of CCP is recommended by the Authors as a feed additive in broiler nutrition.

Response 1: Our CCP sample was extracted from Camellia oleifera seed cake which is a byproduct mainly containing polysaccharides, saponins, and protein. In our study, the extracted samples contained 28.47% polysaccharides, 18.98% protein, 15.00% lignin, 2.13% cellulose, 0.50% ash, and without saponin (Line 80-82). And 200 mg/kg and 800 mg/kg CCP (0.70 g/kg and 2.81 g/kg extracted samples) were added in basis feeding (Line 100-102).

Line 80-82: After major components analysis, the extracted sample contained 28.47% polysaccharides, 18.98% protein, 15.00% lignin, 2.13% cellulose, 0.50% ash, and without saponin.

Line 100-102: In this study, 200 mg/kg and 800 mg/kg CCP (0.70 g/kg and 2.81 g/kg extracted samples) were selected according to Khalaji et al. [28] and Dong et al. [29] and added in broilers basis feeding.

Point 2: Table 1 – shows the base diet. What is the energy and polysaccharide content in experimental mixtures CCP supplemented? Please complete this information.

Response 2: Thanks for your suggestion. 200 mg/kg and 800 mg/kg CCP (0.70 g/kg and 2.81 g/kg extracted samples) were added in basis feeding showing in Line 100-102 instead of Table 1. The metabolizable energy was shown in Table 1. We ignored the energy of the added polysaccharide can supplement when we calculated the energy of diet fed by birds. This type of polysaccharide is constituent of plant cell wall, and is often considered as non-nutritional additive, although a little of this polysaccharide may be digested by birds or gut microorganisms.

Line 100-102: In this study, 200 mg/kg and 800 mg/kg CCP (0.70 g/kg and 2.81 g/kg extracted samples) were selected according to Khalaji et al. [28] and Dong et al. [29] and added in broilers basis feeding.

Point 3: L75- Please provide the approval number of the Ethics Committee for this research.

Response 3: Thank you for reminding us. We supplemented this information in the revised manuscript simultaneously (Line 86-87).

Line 86-87: The approval number of the Ethics Committee for this research is GAASIAS-2018-016.

Point 4: L95 – Why salt, not crude ether, crude fiber and carbohydrates were determined in the mixtures?

Response 4: Thanks for your questions. Admittedly, there were some errors of presentation of nutrient contents. All the nutrient contents in diet including crude protein, calcium and salt were calculated values. We did not analyze the crude ether, crude fiber and carbohydrates in diet. In addition, we revised some inappropriate contents in the Table 1.

Point 5: L146 – To what reference values were the obtained results related? Please complete this information.

Response 5: We might see the different lines in this manuscript. L146 for me is showed in 2.7 blood collection. Wish you mean that give the full names of IgA, IgG, IgM, HDL-C, LDL-C, MDA, NDV Ab, SOD, T-AOC, T-CHO in Line 156-163.

Point 6: L165 – please ANOVA one-way analysis strengthened by polynomial analysis.

Response 6: The trial was a single factorial design and the concentrations of 0, 200 and 800 mg/kg CCP were selected. ANOVA one-way analysis would be enough and polynomial analysis probably needs not to be used.

Point 7: L166 – results are given as average. How many repetitions were there in individual analyzes and measurements?

Response 7: There were 6 replicate pens with 2 or 4 chicks in different analyzes and measurements that has been mentioned in Line 90-91, Line 125-126, Line 152-153, as well as in Table 3-7 (Line 198-199, 224-225, 236-237, 252-253, and 273-274).

Line 90-91: Each treatment had 6 replicate floor pens with 16 chicks (8 hens and 8 cocks) per pen at first day with size of 1.3 × 3.5 m.

Line 125-126: 6 replicate floor pens with 2 chicks (1 hen and 1 cock) per pen at 21, 42 and 50 days of age, representative of the average BW of each group, were selected.

Line 152-153: Before 6 replicate cages per treatment with 4 birds (2 hens and 2 cocks) per cage slaughtered at 50 days, blood was withdrawal from the wing vein.

Line 198-199: Each value is the mean of 6 replicates.

Line 224-225, 236-237, 252-253, and 273-274: Each value is the mean, n=12.

Point 8: L169-170 – this is not information about reporting results. They should be moved to the Discussion chapter.

Response 8: Thanks to reviewer’s comment. They have been moved to 4. Discussion (Line 278-349).

Point 9: L178 – The authors suggest that CCP can improve feed efficiency is too far-reaching. Statistically significant differences were found only in one parameter in the last 7 days of fattening. Supplementation had no significant effect on other CCP parameters. Please provide this information more precisely.

Response 9: Thanks so much to the reviewer’s kindly suggestion. The data in our study about CCP affection could not get the results to improve feed efficiency. Therefore, we tried to express that CCP had the potential to improve feed efficiency (Line 286-287).

Line 286-287: Hence, CCP supplementation had the potential to improve feed efficiency.

Point 10: L187 – Please specify the number of measurements. In Table 3, the authors state that measurements of broiler production parameters were carried out in the amount of n = 12 (1 hen and 1 cock), while in Material and Methods state that the study was carried out at n = 16 (8 hen and 8 cock).

Response 10: In 2.2 Chicks and housing of Material and Methods, it was designed that “Each treatment had 6 replicate floor pens with 16 chicks (8 hens and 8 cocks) per pen at first day” (Line 90-91). However, “at the end of each feeding phase, 6 replicate floor pens with 2 chicks (1 hen and 1 cock) per pen at 21, 42 and 50 days of age, representative of the average BW of each group, were selected, placed in crates overnight, and slaughtered by severing the carotid artery and jugular vein after 16 hours feed withdrawal” (Line 125-128).

16 chicks were put in each pen at first day, and 2 chicks/pen in each feeding phase of 21, 42, and 50 days were sampled and slaughtered. It was easy to cause misunderstanding, so the expression in Table 3-7 has been modified to “Each value is the mean of 6 replicates” (Line 198-199) and “Each value is the mean, n=12” (Line 224-225, 236-237, 252-253, and 273-274).

Line 198-199: Each value is the mean of 6 replicates.

Line 224-225, 236-237, 252-253, and 273-274: Each value is the mean, n=12.

Point 11: L192-194, 196-198, 228-229, 241-244, 277 – please transfer this information to Discussion.

The manuscript is missing the Discussion chapter. This is a serious shortcoming in this publication. The results were presented with a very poor background of basic information. The authors did not even try to translate the results obtained or analyze the processes involved. Without this part, the manuscript cannot be published - it is not of adequate scientific value.

Response 11: Thanks to reviewer’s kindly comment. Some discussion in 3. Results has been moved to 4. Discussion (Line 278-349).

Point 12: Conclusion

L301 – And what were the tests? Not nutritional? Please rewrite Conclusion. What are the most important achievements of using CCP? No recommendations for poultry production. Please provide application forms.

Response 12: Under reviewer’s suggestion, the Conclusion has been revised to focus on the function of the CCP on broilers (Line 354-362).

Line 354-362: Dietary supplementation with Camellia oleifera cake polysaccharides (CCP) had potential function in increasing feed efficiency, and had positive effects on some organ weights and blood profile, without showing any negative effect on growth performance of Lingnan yellow broilers. Besides, CCP exhibited capacity in improving some antioxidant capacity and changing some meat quality attributes. Intestinal probiotics showed significant alterations in response to the CCP supplementation, demonstrating the role of CCP on improving the structure of intestinal flora and favored the intestinal health in Lingnan yellow chicks. Further studies are necessary to define and delineate the role of CCP on metabolism of blood and immune function of gut microbe.
